# Are Bus Company Regulations Associated with Crash Risk? Findings from a Retrospective Survey in Four Chinese Cities

**DOI:** 10.3390/ijerph16081342

**Published:** 2019-04-14

**Authors:** Xiaolin Wu, Huimin Zhang, Wangxin Xiao, Peishan Ning, David C. Schwebel, Guoqing Hu

**Affiliations:** 1Zhou Enlai School of Government, Nankai University, Tianjin 300050, China; wuxiaolin@nankai.edu.cn; 2School of International and Public Affairs, Shanghai Jiao Tong University, Shanghai 200030, China; zhanghuimin@sjtu.edu.cn; 3Department of Epidemiology and Health Statistics, Xiangya School of Public Health, Central South University, Changsha 410078, China; xiaowangxin@csu.edu.cn (W.X.); ningpeishan@csu.edu.cn (P.N.); 4Department of Psychology, University of Alabama at Birmingham, Birmingham, AL 35294, USA; schwebel@uab.edu

**Keywords:** bus crash, bus company regulation, retrospective survey, China

## Abstract

Bus crashes are common in urban China, and bus company regulations are hypothesized to be related to bus crash risk. We conducted a retrospective survey to examine the association in four large Chinese cities (Changsha, Shenzhen, Fuzhou, and Wuhan). Four types of bus crashes were considered: (a) passengers injured while riding the bus; (b) bus colliding with or scraping other motor vehicles; (c) bus colliding with non-motorized vehicles or pedestrians; and (d) bus damaging public facilities. Based on regulations governing the drivers’ work, complete round trips per day, and their paid salary, three categories of companies were studied: type A: ≥14 h worked/day, ≥6 round trips/day, and >70% of salary based on performance; type B: 8–13 h/day, 4 or 5 round trips/day, and 36–70% of salary; and type C: <36% of salary and no other specified requirements. Of the 926 respondents, 20.7% reported one or more crashes or related risk events in the past month. Drivers from the three types of companies reported crash incidence rates of 31.9%, 8.8%, and 6.0%, respectively, in the past month. Type A crash rates were significantly higher than type C after controlling for relevant covariates (adjusted odds ratio (OR) = 7.1, 95% confidence interval (CI): 3.74–13.47). We conclude that more stringent bus company regulations, which mandate drivers to work long hours and obtain salary based on job performance in meeting demanding metrics, are associated with elevated bus-related crash risks. Local governments in China should regulate bus companies to ensure drivers work reasonable hours and are paid based on the quality of their work (e.g., safety).

## 1. Introduction

According to a recent World Health Organization (WHO) report, road traffic injuries are the eighth leading cause of death in the world. Annually, about 1.35 million people die from road traffic crashes, over half of whom (54%) are pedestrians, cyclists, and motorcyclists [1]. As is the case in many countries around the world, buses are a common mode of transportation in urban China. Although buses offer efficiency in urban transportation, they also pose safety risks.

The “Buses Involved in Fatal Accidents” (BIFA) project, conducted in the United States, showed that about 63,000 buses were involved in traffic crashes annually, including 325 that involved a fatal injury, 14,000 with non-fatal injuries, and about 48,000 that resulted in property damage [2]. The 325 fatal crashes caused about 375 fatalities; about 50 of the victims were bus occupants (including drivers), 225 were occupants of other vehicles, and 100 were pedestrians and cyclists [2]. Chinese statistics suggest buses have a higher crash rate than taxis, trucks, tour buses, long-distance bus coaches, or commercial trucks with trailers [3].

Bus crash rates in China are currently high. As an example, police statistics from Lanzhou show that bus-involved crashes accounted for 10% of the total number of road traffic deaths and injuries from 2011 to 2014 [3]. Scholarly research also indicates buses frequently violate traffic laws; reports from both Guangzhou (traffic law violation rate of 36.9%) [4] and Changsha (violation rate of 20.2% in 2015) [5] demonstrate the frequency of violations.

Bus-related crashes are concerning because they pose risk to two groups, namely, passengers and other road users. Because buses often carry many passengers, risk can be amplified: a serious crash can have a magnified impact on morbidity and mortality. Buses also pose significant risk to other road users and to roadside facilities given their size and weight, which can inflict serious injury on pedestrians, cyclists, and other vulnerable road users who are struck, and can damage public transportation facilities [6,7,8,9,10].

Given this, in addition to data on crash and violation rates, bus safety has emerged as a priority for urban road traffic injury prevention efforts in China and elsewhere. Many local governments and enterprises have hired full-time bus safety officers [11] and implemented routine field inspections [12] to improve the safety of urban bus service. Previous research indicates that male drivers [13], drivers with a driving history of 6–20 years rather than less or more, roads allowing a high driving speed or having commercial blocks, lack of bus-first traffic control designs [14], outdated vehicles, and autumn, winter, and weekend time [5] are associated with higher crash risks.

Related research with other professional drivers supports these findings. A retrospective survey in Iran, for example, showed that taxi drivers—who often travel on urban roads and in heavy traffic—are more likely to commit errors (e.g., ordinary and aggressive violations) than truck drivers, who probably travel more often in less trafficked areas [15]. Similarly, a study of professional drivers in Colombia found that 29.1% suffered from high job strain and that the job-related stress was associated not just with drivers’ mental health, but also with their rate of traffic crashes and fines [16]. A second study of professional drivers in Colombia (57.4% city bus, 17.6% taxi, 25% long-distance bus drivers) concluded that traffic penalties to professional drivers are related to work-related factors such as stress, and that complex relationships exist between work-related driving stress, individual differences, and traffic penalties [17].

One factor that has not yet been examined in the published literature is the role of regulations governing professional drivers on crash risk. Harsh and unreasonable regulations for drivers, such as those commonly in place for urban bus drivers in China, could lead to risky driving behaviors and violations of the law. Such harsh regulations are common in China because bus companies operate through a public–private partnership, and each company is permitted to stipulate their own driver regulations. Financially, bus companies operate through a combination of government subsidies and low-cost ticket prices; local governments subsidize buses to enable their citizens to travel conveniently and cheaply throughout the city. However, because bus companies are privately owned, they maintain profit motivation and therefore may set regulations that increase profit but impact safety. Some companies mandate, for example, that drivers complete a certain number of round-trip routes per day or that they complete their routes in a timely manner despite facing heavy traffic conditions. Failure to complete such mandates reduces driver salaries [18].

In this study, therefore, we examined associations between bus company regulations and bus crash risks in China. We hypothesized that harsher bus company regulations would motivate drivers to drive dangerously, engaging in risky behaviors like drowsy or distracted driving and/or violating laws like speeding and running red lights [19]. The hypotheses were tested using a retrospective survey with a large sample of bus drivers in four major Chinese cities.

## 2. Methods 

### 2.1. Design

A self-report cross-sectional survey examined associations between recent bus crash incidence and bus company management policies in four Chinese cities. The survey was completed by 926 professional bus drivers working in urban settings.

### 2.2. Sample

We included four large Chinese cities in our research—Changsha, Fuzhou, Shenzhen, and Wuhan—because each city had multiple bus companies with regulations that offer their drivers performance-based salary payments under different formulas.

Changsha is the capital of Hunan Province and had a total population of 7.92 million people in 2017 [20]. Changsha has adopted a market-based model for bus operation, with nine bus companies in operation. All bus companies in Changsha implement highly similar regulations with extremely strict requirements on daily driving trips. In almost all cases, bus driver salaries are linked to the completion of objective job performance tasks. We selected the largest bus company in Changsha to survey.

Fuzhou is the capital of Fujian Province, with a total population of 6.93 million in 2017 [21]. Fuzhou has fewer bus companies than Changsha, and we selected the two largest companies to survey. Shenzhen is located in the south of Guangdong Province and had a population of 12.53 million citizens in 2017 [22]. The Shenzhen municipal government integrates 38 bus companies into three large companies. We surveyed the largest bus company.

Wuhan is the capital of Hubei Province and had a population of 10.89 million in 2017 [23]. Unlike the first three cities, whose bus services are delivered only by private bus companies, Wuhan Public Transport Group Corporation Limited is owned by the Wuhan municipal government. It has 330 bus lines and the length of the line is 6005.9 km. We selected five companies (including three state-owned subsidiaries and two public–private joint ventures) to survey.

Thus, a total of nine bus companies were surveyed from the four cities. Because no previous research offered guidance to estimate effect sizes for power analysis, we conservatively planned to survey at least 200 bus drivers in each of the four cities.

### 2.3. Bus Crash Risk Measure

Based on previous literature [6,7,8,9,10], we assessed four outcomes as proxies for bus crash risk: (a) passengers injured while riding the bus; (b) bus colliding with or scraping other motor vehicles; (c) bus colliding with non-motorized vehicles or pedestrians; and (d) bus damaging public facilities. Each outcome was assessed through drivers’ reports about occurrence in the past month prior to completing the survey.

### 2.4. Performance-Based Salary Payment

The Chinese Code for Transport Planning on Urban Road dictates that the length of round-trip urban bus circuits should be 8–12 km [24]. Although the precise distance for bus circuits varies, typical operation times for a complete circuit are similar, averaging about 2 h. For our research, we conducted interviews with bus drivers and managers (detailed below) to gather information and then divide the nine participating bus companies into three categories based on their regulations for minimal driving hours and trips per weekday for drivers, and the proportion of driver salary linked to the completion of the assigned tasks:

Type A: Companies in this category required bus drivers to drive at least 14 h per day, to make six round trips of their route each day, and they paid over 70% of drivers’ salary according to drivers’ performance to complete designated job performance tasks.

Type B: Companies in this category required bus drivers to drive 8–13 h per day, to make four or five round trips each day, and they paid 36–70% of drivers’ salary according to drivers’ performance.

Type C: Companies in this category required drivers to work for no more than 8 h per day, to make fewer than four round trips per day (usually three trips are required), and they paid less than 36% of drivers’ salary according to job performance.

### 2.5. Covariates

We considered three self-reported driver characteristics as covariates: sex, level of education (≤9 years, 10–12 years, or ≥13 years), and years working as a bus driver (<5 years, 5–9 years, 10–14 years, or ≥15 years).

### 2.6. Data Collection

Following informed consent processes, trained researchers stood outside bus control points (small rooms located at the starting points for bus lines) in Changsha, Shenzhen, and Fuzhou, and approached bus drivers during their brief (5–8 min) rest between routes to invite them to participate in the study. (Note: Bus control points are also called dispatch rooms. They typically are small rooms located at the starting point of bus service routes. Since there are many bus routes in China’s large cities, bus companies are required to have a dispatch room at the starting point for each bus route. The staff members working in the dispatch room remind bus drivers of their departure time and record the departure and arrival time of each bus every day. Dispatch rooms also offer a location for bus drivers to rest briefly between routes.) All invited drivers consented to participate and trained data collectors completed the interview-based survey, which took 5–8 min with each driver. Data collection in Wuhan was conducted primarily at a bus driver conference (*n* = 179) and was supplemented with dispatch room methods similar to the other three cities (*n* = 21). Surveys using the dispatch room method in Wuhan were conducted after the conference and excluded drivers who had previously participated in the survey.

We also used a structured oral interview with managers of each bus company to collect information concerning how bus driver salaries were determined and the extent to which job performance metrics were used to determine salary, as well as basic descriptive characteristics of the company (e.g., number of bus lines the company operated, number of bus drivers employed). All survey questions were developed, pilot-tested, and refined prior to implementation.

### 2.7. Ethical Statement

Informed consent was provided by all participants and all data were collected anonymously. The research protocol was approved by the Medical Ethics Committee of Central South University (No. XYGW-2017-54). Data collection was completed between 1 August 2017 and 30 October 2017.

### 2.8. Statistical Analysis

The incidence of the four bus-related crash risk outcomes and 95% confidence intervals (95% CI) were calculated first to estimate bus crash risks that drivers reported encountering in the past month. Next, multivariate logistic regression was conducted to examine associations between bus-related crash risk outcomes and the performance-based payment method used by the companies, after adjusting for covariates (gender, level of education, and years working as a bus driver). Crude and adjusted odds ratio (OR) rates were calculated to quantify the associations. SPSS 18.0 (IBM, Armonk, New York, United States) was used to perform statistical analyses. *p* < 0.05 was considered to be statistically significant.

## 3. Results

### 3.1. Sample Characteristics

In total, we collected 926 valid questionnaires from bus drivers (Table 1). Men constituted 94.8% of the 926 respondents. Of the surveyed drivers, 56.4% reported 10–12 years of formal education, followed by 32.3% who reported nine or fewer years of formal education and 11.3% who had at least some college education. Of the drivers, 36.8% had 5–9 years of experience working as bus driver, and 25.8% had 10–14 years of experience. Of the 926 drivers, 282 (30.5%) worked at a type A company, 444 (47.9%) at a type B company, and 200 (21.6%) at a type C company. The gender, education, and experience variables all differed significantly across the types of bus companies, *p* < 0.05, with type C companies generally hiring more female, better educated, and more experienced drivers.

### 3.2. Bus-Related Crash Incidents

Of the 926 drivers, 20.7% reported one or more crash or crash-risk events in the past month. Among them, 141 (15.2% of full sample, 95% CI: 12.9–17.7%) reported just one crash or crash-risk event in the past month, 43 (4.6%, 95% CI: 3.3–6.2%) reported two crash or crash-risk events, and a handful of drivers (0.9%, 95% CI: 0.3–1.5%) experienced three or more crash events (Table 2). The most frequently reported bus crash event was “colliding with or scraping other motor vehicles” (12.1% of drivers; 95% CI: 10.0–14.3%), followed by “passengers injured while riding the bus” (6.2% of drivers; 95% CI: 4.6–7.9%), “colliding with non-motorized vehicles or pedestrians” (1.5% of drivers; 95% CI: 0.8–2.3%), and “damaging public facilities” (1.2% of drivers; 95% CI: 0.5–1.9%).

### 3.3. Associations between Bus-Related Crash Incidents and Performance-Based Salary Payment

For the logistic regression models, we combined the four crash risk measures into a single dichotomous indicator, whether the driver had experienced a crash in the past month or not. The models showed significant associations between bus crash events and performance-based salary payments at the company level. Compared to bus drivers from companies with less restrictive work requirements and a lower proportion of salary payment based on performance (type C), those from companies with the most restrictive work requirements and highest degree of salary based on job performance (type A) had a much higher crash incidence (adjusted OR: 7.10, 95% CI: 3.74–13.47), with unadjusted crash incidence of 6.0% vs. 31.9% (Table 3). Other associations were not statistically significant.

## 4. Discussion

### 4.1. Primary Findings

This study examined associations between urban bus company regulations and bus crash risks in China. Our findings support the hypothesis that harsher bus company regulations, which mandate drivers to work long hours and obtain salary based on job performance to meet demanding metrics, are associated with elevated bus-related crash incidence. In addition, we found that even among drivers from bus companies having the laxest regulations, bus-related crash incidence in the past month was 6.0% (compared to 8.8% and 31.9% incidence rates among drivers from companies having moderate and strict regulations).

Previous research indicates bus crashes are frequent in China, and that the contribution of buses to traffic crashes and to traffic-related injury exceeds that of other types of vehicles [25]. Our findings concerning one-month crash incidence rates (6.0–31.9% across the types of companies studied) support such assertions. Our results also provide evidence to suggest that the environment faced by drivers, both from their company regulations and from the road environment in China [26], may contribute to crash risks. Our findings are in agreement with previous reports that regulations encouraging truck driver risk-taking, such as incentives for working long hours and completing routes quickly, lead to crash risk [27]. Although significant associations were not detected between bus crash risks with gender, level of education, and working years as a bus driver in this study, the adjusted odds ratios substantially deviated from 1.0, suggesting the effects of inadequate sample size to detect small but potentially meaningful effects. Although not demonstrated empirically in our study, other risks may also occur among inappropriately incentivized bus drivers, such as speeding, risky lane-changing, running red lights, and drowsy driving.

The public health solutions to the problems our data illustrate are complicated. Bus companies in China are primarily driven by profit. Without rigorous government supervision, bus companies are incentivized to implement unreasonable regulations that enable them to increase their profits, even if those regulations may elevate crash risks. Ultimately, each involved party—bus companies, passengers, and the government—have inconsistent objectives that need to be reconciled with safety as a common priority. Bus companies desire efficient and timely completion of routes, some degree of customer service, high passenger fares, and low employee salaries. Passengers prioritize staying safe, getting to their destination on time, and paying low fares for their bus tickets. Government authorities focus on safe, efficient, and timely transportation for their citizens. Balancing the conflicting considerations of these three groups is complicated, but ultimately government regulations may yield the safest and most efficient results. Government bodies must enhance supervision and regulation of companies to prohibit unreasonable regulations that incentivize bus drivers to engage in risky practices. In fact, some current company regulations violate existing Chinese labor laws by requiring drivers to work over 8 h per day [28], but those laws are not enforced. In conducting oversight of companies, however, government officials must also balance the profit needs of bus companies and the fact that the cost of bus tickets must remain low to ensure the basic transportation rights of all Chinese citizens [29]. Increased government subsidies to bus companies might help the companies manage low-cost bus services while still providing safe, efficient, and low-cost bus service for all citizens.

### 4.2. Limitations

Our research has several limitations. First, our sample was limited to just four Chinese cities and 926 participants. The results are not representative of all Chinese cities, although they reflect the situation across much of China. According to Urban Public Transport Regulations [30], bus management regulations in almost all major Chinese cities adopt regulations similar to the three categories we considered in this study. Furthermore, the sample size is inadequate to detect weak or moderate associations, such as crash incidence differences that may exist between male and female drivers. Second, we did not collect the information of individual characteristics in our survey, and thus cannot adjust for their impact on our results. Drivers of most bus companies in China are similar in terms of sex and age, as demonstrated by the data in Table 1, but we did not collect data on driver personality or temperament. Finally, we relied on self-report survey methods to collect our data. Such surveys are vulnerable to reporting biases; in particular, some bus drivers may not want to report crash events they experienced. We worked to limit this bias by using anonymous surveys and collecting no private information from the surveyed drivers, but future research might consider strategies to collect crash data from police or bus company records rather than driver self-report.

## 5. Conclusions

Bus company regulations that request drivers to work long hours and that incentivize driver salaries for their performance were associated with high bus crash incidence rates in urban China. The results highlight the need for government authorities to regulate bus companies and ensure public safety through inspection, management, legislation, and regulation of bus driver work duties. Specifically, the government should forbid bus companies from placing unrealistic or dangerous requirements on bus drivers, such as driving for long periods of time and linking salary benefits to job performance metrics. Governmental subsidies to bus companies should continue to help bus companies achieve reasonable profit while still delivering safe and affordable bus services.

## Figures and Tables

**Table 1 ijerph-16-01342-t001:** Characteristics of surveyed bus drivers.

Variable	*N*	Percent (%)	Salary Payment Method (%)			χ^2^	*p*
			Type A	Type B	Type C		
Total	926	100	30.5	47.9	21.6		
Gender						6.741	0.034
Male	878	94.8	94.7	96.4	91.5		
Female	48	5.2	5.3	3.6	8.5		
Years of education						98.261	<0.01
≤9 years	355	38.3	32.3	47.3	27.0		
10–12 years	475	51.3	56.4	50.2	46.5		
≥13 years	96	10.4	11.3	2.5	26.5		
Years working as bus driver						3.063	0.801
<5 years	191	20.6	21.3	19.6	22		
5–9 years	341	36.8	35.1	37.8	37		
10–14 years	239	25.8	28.7	24.8	24		
≥15 years	155	16.7	14.9	17.8	17		

Note: Bus companies were divided into three categories based on required driver working hours, round trips each workday, and proportion of salary based on the completion of required tasks: (a) type A: drivers work ≥14 h/day, complete ≥6 round trips/day, and are paid >70% of salary based on performance (b) type B: drivers work 8–13 h/day, complete 4 or 5 round trips/day, and are paid 36–70% of salary based on performance; (c) type C: drivers have no specified requirements for driving hours or round trips per day and are paid <36% of their salary based on performance.

**Table 2 ijerph-16-01342-t002:** Bus driver report of traffic crashes in the past month.

Variable	*N*	Proportion % (95% CI)
Type of crash		
Passengers injured while riding the bus	57	6.2 (4.6, 7.9)
Colliding with or scraping other motor vehicles	112	12.1 (10.0, 14.3)
Colliding with non-motorized vehicles or pedestrians	14	1.5 (0.8, 2.3)
Damaging public facilities	11	1.2 (0.5, 1.9)
Number of crashes		
1	141	15.2 (12.9, 17.7)
2	43	4.6 (3.3, 6.2)
≥3	8	0.9 (0.3, 1.5)

**Table 3 ijerph-16-01342-t003:** Associations between self-reported bus crashes in the past month, type of bus company, and driver demographics.

Variables	Incidence Rate		Crude OR (95% CI)	Adjusted OR (95% CI)
	Proportion (%)	95% CI		
Performance-based salary payment				
Type A	31.90	(26.44, 37.39)	7.34 (3.89, 13.86) *	7.10 (3.74, 13.47) *
Type B	8.80	(6.14, 11.43)	1.51 (0.77, 2.95)	1.41 (0.71, 2.80)
Type C (Ref.)	6.00	(2.68, 9.32)		
Gender				
Male	15.40	(13.05, 17.83)	1.58 (0.61, 4.05)	1.62 (0.60, 4.35)
Female (Ref.)	11.10	(1.56, 20.66)		
Level of education				
≤9 years	13.90	(10.17, 17.56)	1.15 (0.58, 2.25)	1.25 (0.60, 2.60)
10–12 years	16.70	(13.42, 20.05)	1.40 (0.73, 2.68)	1.33 (0.67, 2.69)
≥13 years (Ref.)	12.40	(5.70, 19.04)		
Working years as bus driver				
<5 years	15.18	(10.05, 20.32)	0.98 (0.54, 1.76)	0.93 (0.40, 1.73)
5–9 years	15.54	(11.68, 19.41)	1.00 (0.59, 1.70)	0.98 (0.56, 1.70)
10–14 years	14.64	(10.13, 19.16)	0.94 (0.53, 1.65)	0.80 (0.44, 1.45)
≥15 years (Ref.)	15.48	(9.73, 21.24)		

* *p* < 0.05. Note: Bus companies were divided into three categories based on required driver working hours, round trips each workday, and proportion of salary based on the completion of required tasks: (a) type A: drivers work ≥14 h/day, complete ≥6 round trips/day, and are paid >70% of salary based on performance (b) type B: drivers work 8–13 h/day, complete 4 or 5 round trips/day, and are paid 36–70% of salary based on performance; (c) type C: drivers have no specified requirements for driving hours or round trips per day and are paid <36% of their salary based on performance.

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
