# Peer review of "Are Bus Company Regulations Associated with Crash Risk? Findings from a Retrospective Survey in Four Chinese Cities"

_ijerph, 2019, doi:10.3390/ijerph16081342_

Round 1

Reviewer 1 Report

Although the topic of this research is interesting, the study design and the analysis carried out in this study are flawed. As a quick example, this study was based on self-reported surveys, and the dependent variable was whether a respondent was involved in a crash in the previous month. Overall, the validity of this research is questionable, and I do not recommend this research for publication. 

Author Response

Point 1: Although the topic of this research is interesting, the study design and the analysis carried out in this study are flawed. As a quick example, this study was based on self-reported surveys, and the dependent variable was whether a respondent was involved in a crash in the previous month. Overall, the validity of this research is questionable, and I do not recommend this research for publication.

Response 1: Thanks. We agree with the reviewer that self-report has its limitations, but in novel domains of research, self-report data is commonly used, especially when alternative approaches are unrealistic, prohibitively expensive, or potentially invalid. That is the case for some of the data we collected in this study. We now address the limitation of self-report methods in the revised discussion. See below for a listing of example articles using self-report methodology techniques published in well-respected academic journals.

1. Schneider ALC, Wang D, Ling G, Gottesman RF, Selvin E. Prevalence of Self-Reported Head Injury in the United States. N Engl J Med. 2018 Sep 20;379(12):1176-1178. doi: 10.1056/NEJMc1808550. PubMed PMID: 30231228; PubMed Central PMCID: PMC6252182.

2. Suarez FL, Savaiano DA, Levitt MD. A comparison of symptoms after the consumption of milk or lactose-hydrolyzed milk by people with self-reported severe lactose intolerance. N Engl J Med. 1995 Jul 6;333(1):1-4. PubMed PMID: 7776987.

3. Colver A, Rapp M, Eisemann N, Ehlinger V, Thyen U, Dickinson HO, Parkes J, Parkinson K, Nystrand M, Fauconnier J, Marcelli M, Michelsen SI, Arnaud C. Self-reported quality of life of adolescents with cerebral palsy: a cross-sectional and longitudinal analysis. Lancet. 2015 Feb 21;385(9969):705-16. doi: 10.1016/S0140-6736(14)61229-0. Epub 2014 Oct 7. PubMed PMID: 25301503;

PubMed Central PMCID: PMC4606972.

4. Dickinson HO, Parkinson KN, Ravens-Sieberer U, Schirripa G, Thyen U, Arnaud C, Beckung E, Fauconnier J, McManus V, Michelsen SI, Parkes J, Colver AF. Self-reported quality of life of 8-12-year-old children with cerebral palsy: a cross-sectional European study. Lancet. 2007 Jun 30;369(9580):2171-2178. doi: 10.1016/S0140-6736(07)61013-7. PubMed PMID: 17604799.

Reviewer 2 Report

1. You may improve the review of the literature with results os similar studies in other countries.

2. The description of the procedure for the data collection in the different cities needs to be improved. The dipatch room method needs to be described. I did not get a clear sense of what was done.

2. The categories to divide the bus companies and make the comparisson used information about hours per day of work, number of round trips AND proportion of driver salary linked to completion of the assigned task. However, in many points of the paper you refer to this criteria as performance-based salary payment. If I understand properly, this is just one of the criteria to create the categories. This confusion needs to be corrected.

Author Response

Point 1: You may improve the review of the literature with results os similar studies in other countries.

Response 1: Thanks. We have integrated relevant studies in the background to improve the manuscript. Please see changes in the revised manuscript.

Point 2: The description of the procedure for the data collection in the different cities needs to be improved. The dispatch room method needs to be described. I did not get a clear sense of what was done.

Response 2: Thanks for the suggestion. We have detailed the dispatch room in the methods section. The dispatch room is the bus control point, which typically means a small room located at the starting point of bus service route. There are many bus routes in China's large cites, and bus companies are required to have a dispatch room at the starting point for each bus route. The staff members working in the dispatch room remind bus drivers of their departure time and record the departure and arrival time of each bus every day. They also recharge bus cards for passengers and provide passengers with invoices. Dispatch rooms also offer a location for bus drivers to rest between routes. For this study, our research team waited outside dispatch rooms to interview bus drivers while they rested (note that the interview was short, lasting 5-8 minutes, and did not affect the drivers’ work). Please see changes in the revised manuscript.

Point 3: The categories to divide the bus companies and make the comparison used information about hours per day of work, number of round trips AND proportion of driver salary linked to completion of the assigned task. However, in many points of the paper you refer to this criteria as performance-based salary payment. If I understand properly, this is just one of the criteria to create the categories. This confusion needs to be corrected.

Response 3: Many thanks. We have corrected the wording to clarify this issue throughout the manuscript. Please see changes in the revised manuscript.

Reviewer 3 Report

The manuscript is, overall, sound and pertinent to address a very serious problem of occupational and public health that affects the industry of transportation worldwide.

The abstract is adequate and summarizes the contents adequately. However, the conclusion to present in this section should be more based on the findings of the study (also emphasizing on the work schedules and environment), apart from its implications.

The background of the study is, perhaps, the biggest gap of the manuscript. More information presenting and relating the study variables and its influence on occupational accidents of bus drivers should be described in page 2, second paragraph. I suggest the authors to address task conditions, fatigue, stress and health outcomes (undisputedly related to the research problem) in it, as a manner of presenting to the readers a more holistic context on the adverse conditions of professional/commercial driving. I suggest the authors to refer to (e.g.); DOIs 10.7717/peerj.6249; 10.1016/j.trf.2018.12.010; 10.3390/ijerph15030497. This because it is still a bit unclear the human-based mechanism that may explain how these crashes take place and, thus, could be prevented through the evidence-based intervention.

Also, the hypothesis (lines 31 to 35) should be presented just after the aim of the study.

Methods are adequate, but some minor adjustments should be done:

-       To point the type of study (cross-sectional and self-report-based research?) in the section 2.1.

-       In 2.8 (statistical analysis) it is important to describe a bit more the aim of the analyses, specially multivariates (for readers).

Results are adequately described, and Tables support the data very well.

As for discussion, it is convenient to remind the readers the aim of the study before the hypothesized results.

Also, there is a set of outstanding study outcomes (e.g., some key descriptive results and associations between driver profiles and self-reported crashes) that should be more discussed in the glance of the existing literature in the field. Also, it is suggestible to cite papers linking demographic variables and risky road behaviors explaining a large percentage of negative safety outcomes.

Further, and although the main conclusion is majorly based on the actual data, the second part (implications and policymaking) could be better supported in the last paragraph of the discussion.

Finally, I believe there are some key limitations that should be acknowledged: the possibility of CMB (common method biases) directly related to the data collection method, and the lack of specificity in crash severity. Perhaps, mentioning these two issues authors could increase the interpretative rigor of the paper and suggest relevant guidelines for further studies in the field.

* As a side comment, please check the writing for the second time: there are some minor grammar errors and typos along the manuscript.

Author Response

Point 1: The manuscript is, overall, sound and pertinent to address a very serious problem of occupational and public health that affects the industry of transportation worldwide.

The abstract is adequate and summarizes the contents adequately. However, the conclusion to present in this section should be more based on the findings of the study (also emphasizing on the work schedules and environment), apart from its implications.

Response 1: Many thanks. We have improved the conclusion as suggested. Please see changes in the revised manuscript.

Point 2: The background of the study is, perhaps, the biggest gap of the manuscript. More information presenting and relating the study variables and its influence on occupational accidents of bus drivers should be described in page 2, second paragraph. I suggest the authors to address task conditions, fatigue, stress and health outcomes (undisputedly related to the research problem) in it, as a manner of presenting to the readers a more holistic context on the adverse conditions of professional/commercial driving. I suggest the authors to refer to (e.g.); DOIs 10.7717/peerj.6249; 10.1016/j.trf.2018.12.010; 10.3390/ijerph15030497. This because it is still a bit unclear the human-based mechanism that may explain how these crashes take place and, thus, could be prevented through the evidence-based intervention.

Response 2: Thanks for the great comment. We have adopted it and improved the background as suggested. Please see changes in the revised manuscript.

Point 3: Also, the hypothesis (lines 31 to 35) should be presented just after the aim of the study.

Response 3: Many thanks. We have added the purpose of the study before the research hypothesis. Please see changes in the revised manuscript.

Point 4: Methods are adequate, but some minor adjustments should be done:

-To point the type of study (cross-sectional and self-report-based research?) in the section 2.1.

-In 2.8 (statistical analysis) it is important to describe a bit more the aim of the analyses, specially multivariates (for readers).

Response 4: Thanks. We have adopted the suggestions and improved the wording of methods. Please see changes in the revised manuscript.

Point 5: Results are adequately described, and Tables support the data very well.

As for discussion, it is convenient to remind the readers the aim of the study before the hypothesized results.

Response 5: Many thanks for the good point. We now mention the primary aim of this study in the beginning of the discussion. Please see changes in the revised manuscript.

Point 6: Also, there is a set of outstanding study outcomes (e.g., some key descriptive results and associations between driver profiles and self-reported crashes) that should be more discussed in the glance of the existing literature in the field. Also, it is suggestible to cite papers linking demographic variables and risky road behaviors explaining a large percentage of negative safety outcomes.

Response 6: Thanks. As suggested, we have added such discussions on outcome differences across demographic variables and risky road behaviors. Please see changes in the revised manuscript.

Point 7: Further, and although the main conclusion is majorly based on the actual data, the second part (implications and policymaking) could be better supported in the last paragraph of the discussion.

Response 7: Thanks. We have improved the second part of the discussion, as suggested. Please see changes in the revised manuscript.

Point 8: Finally, I believe there are some key limitations that should be acknowledged: the possibility of CMB (common method biases) directly related to the data collection method, and the lack of specificity in crash severity. Perhaps, mentioning these two issues authors could increase the interpretative rigor of the paper and suggest relevant guidelines for further studies in the field.

Response 8: Many thanks. As suggested, we have included these issues in the limitations section. Please see changes in the revised manuscript.

Point 9: * As a side comment, please check the writing for the second time: there are some minor grammar errors and typos along the manuscript.

Response 9: Many thanks. We have double checked the manuscript and corrected grammar errors.
